# Cancer Related Anemia: An Integrated Multitarget Approach and Lifestyle Interventions

**DOI:** 10.3390/nu13020482

**Published:** 2021-02-01

**Authors:** Valentina Natalucci, Edy Virgili, Federica Calcagnoli, Giacomo Valli, Deborah Agostini, Sabrina Donati Zeppa, Elena Barbieri, Rita Emili

**Affiliations:** 1Department of Biomolecular Sciences, University of Urbino Carlo Bo, 61029 Urbino, Italy; valentina.natalucci@uniurb.it (V.N.); deborah.agostini@uniurb.it (D.A.); 2School of Biosciences and Veterinary Medicine, University of Camerino, 62032 Camerino, Italy; virgiliedy@gmail.com (E.V.); federica.calcagnoli@gmail.com (F.C.); 3Neuromuscular Physiology Laboratory, Department of Biomedical Sciences, University of Padova, 35131 Padova, Italy; giacomo.valli@phd.unipd.it; 4U.O.C. Oncologia Medica, ASUR Area Vasta 1, Ospedale Santa Maria della Misericordia di Urbino, 61029 Urbino, Italy; emilirita@gmail.com

**Keywords:** cancer-related anemia, iron supplementation, hepcidin, gut microbiota, lifestyle

## Abstract

Cancer is often accompanied by worsening of the patient’s iron profile, and the resulting anemia could be a factor that negatively impacts antineoplastic treatment efficacy and patient survival. The first line of therapy is usually based on oral or intravenous iron supplementation; however, many patients remain anemic and do not respond. The key might lie in the pathogenesis of the anemia itself. Cancer-related anemia (CRA) is characterized by a decreased circulating serum iron concentration and transferrin saturation despite ample iron stores, pointing to a more complex problem related to iron homeostatic regulation and additional factors such as chronic inflammatory status. This review explores our current understanding of iron homeostasis in cancer, shedding light on the modulatory role of hepcidin in intestinal iron absorption, iron recycling, mobilization from liver deposits, and inducible regulators by infections and inflammation. The underlying relationship between CRA and systemic low-grade inflammation will be discussed, and an integrated multitarget approach based on nutrition and exercise to improve iron utilization by reducing low-grade inflammation, modulating the immune response, and supporting antioxidant mechanisms will also be proposed. Indeed, a Mediterranean-based diet, nutritional supplements and exercise are suggested as potential individualized strategies and as a complementary approach to conventional CRA therapy.

## 1. Introduction

Cancer-related anemia (CRA) is a common comorbidity in cancer patients at diagnosis [1,2]. It does not appear to be a consequence of concurrent antineoplastic therapy; rather it is primarily caused by the low-grade chronic inflammation associated with cancer [3]. Indeed, the biological and hematologic characteristics are similar to those described in anemia associated with other chronic inflammatory pathologies [3]. In particular, CRA is associated with elevated inflammatory markers, and it is characterized as a mild, normocytic normochromic anemia with hemoglobin ranging from 8 to 10 g/dL and a decreased circulating serum iron concentration and transferrin saturation despite ample iron stores (e.g., serum ferritin >100 ng/mL). This suggests that the underlying mechanism of CRA is a defect in iron handling rather than a lack of iron per se. Such a condition is termed “functional iron deficiency” [1].

The pathogenesis of CRA is complex and, in most cases involves a range of factors related to cancer-induced chronic activation of the immune response, resulting from direct and indirect suppressing effects of cytokines in erythropoiesis. Moreover, the release of pro-inflammatory cytokines in cancer patients is frequently associated with increases in reactive oxygen species (ROS), either as a component of the immune response or as a consequence of increased metabolism [4]. ROS may inhibit erythropoiesis [5], interfere with nutritional status and exacerbate the anemia.

Indeed, a strong correlation has been found between the prevalence and severity of anemia, the prevalence of increased mean plasma levels of inflammation markers-including C-reactive protein (CRP), fibrinogen, interleukin (IL)-6, tumor necrosis factor (TNF)-α, IL-1β, erythrocyte sedimentation rate, ferritin, hepcidin, erythropoietin, and reactive oxygen species—and the stage of cancer [3,6,7].

In addition, malnutrition, lack of specific components (such as iron, vitamins, folic acid, etc.) fundamental for erythropoiesis, eating disorders, inactivity and other lifestyle factors may contribute to the multifactorial pathogenesis of anemia [1].

To better understand the prognosis and to properly manage anemic cancer patients it is therefore crucial to correctly identify the existing low chronic inflammatory and oxidative state and to take into consideration the diagnosis of anemia before initiating antineoplastic treatment. An appropriate treatment strategy for CRA patients should strive to target the factors causing the anemia.

In this review, particular attention will be focused on lifestyle interventions in CRA management, since nutrition and exercise might influence the prognosis of this disease and its complications. To help clinicians to provide safe and effective treatments, the relationship between diet, exercise, gut microbiota, and the patient’s general state of health will be analyzed along with the use of anti-inflammatory and anti-oxidative supplements, which are known to have beneficial effects on cancer management.

## 2. Clinical Relevance and Features of CRA

CRA occurs in more than 30% of cancer patients before the initiation of anticancer therapy [1], while published data from the European Cancer Anemia Survey (ECAS) suggest that, among non-anemic cancer patients at the start of anti-tumor therapy, the incidence of anemia after chemotherapy is 63%, after chemo-radiotherapy 40%, and after radiotherapy 20% [8]. The prevalence seems to increase with age [9] and might differ according to the cancer typology. Indeed, the highest percentage of anemic patients is reported in lung, gynecologic or genitourinary, and gastrointestinal cancers [8,9,10]. CRA may accompany the evolution of the cancer, and it is commonly identified in patients at advanced cancer stages [8,10,11,12]. In addition, there is an increasing body of evidence suggesting that anemia is an independent factor adversely affecting antineoplastic treatment efficacy and patient survival [13].

As mentioned above, during its development, the neoplastic pathology leads to immunological alterations due to the interaction between the cancer and the patient’s immune system [14]. When physiological compensatory mechanisms fail to restore homeostasis, various anemia-related clinical symptoms may occur. Some of the most common symptoms, including palpitations, tachycardia, pallor, and cold skin, contribute to cancer-related fatigue, one of the core clinical features in patients living with and beyond cancer. On the other hand, other symptoms are more directly related to the reduced capacity of erythrocytes to transport O_2_ around the body and therefore affect the metabolic activities and functional specificities of organ systems. Changes in cellular metabolism underlie the signs of anemia that may compromise the psychophysical well-being of patients. The effects of anemia on the pulmonary and cardiovascular systems, as well as on the skeletal muscle, can be manifest as physical work capacity and exercise tolerance reduction, resulting in breathlessness, tiredness, and muscle fatigue [15]. In addition, reduced perfusion of the intestinal tract due to shunting of blood to vital organs may cause anorexia, nausea, and malabsorption, which together may negatively impact the patient’s nutritional status and energy levels [16,17]. Interestingly, this range of anemia-induced physiological, endocrine, and metabolic alterations are mechanisms by which the body tries to counterbalance the effects of anemia [17].

## 3. Iron Homeostasis

Iron is a micronutrient and an essential component of human metabolism. Generally, plasma iron levels are maintained by the intestinal absorption and storage of iron. Iron homeostasis is ensured by hepcidin, a recently characterized hormone of the endocrine system. Hepcidin is a cationic amphipathic peptide produced in the liver, released in plasma, and excreted in the urine. It is the homeostatic regulator of intestinal iron absorption and storage, iron recycling by macrophages that phagocytize old erythrocytes, and iron mobilization from liver deposits [18,19], but it is also strongly induced during infections and inflammation [20]. Iron homeostasis is necessary because its deficiency leads to anemia, whereas its excess induces oxidative stress, causing inflammation, cell apoptosis or necrosis, and system dysfunction, including cancer.

Under the influence of hepcidin, macrophages, hepatocytes, and enterocytes retain iron that would otherwise be released into the plasma. Hepcidin works by inhibiting iron outflow through ferroportin, the only known transmembrane iron exporter that is expressed on enterocytes, macrophages, and hepatocytes [19,21]. Hepcidin synthesis is increased by iron loading and decreased by anemia and hypoxia [22]. Increased circulating hepcidin leads to ubiquitination, endocytosis, and degradation of ferroportin in cells involved in iron metabolism, such as macrophages, placental cells, hepatocytes, duodenal cells, etc. [23]. This leads to a reduction in erythropoiesis, as iron is retained inside cells and not made available to erythroid precursors, while cellular ferritin increases, leading in turn to an increase in serum ferritin to maintain the equilibrium inside/outside the cells.

Recently, hepcidin has been found to coordinate iron homeostasis engaging with erythroferrone, another hormone generated by erythropoietin-activated erythroid precursors in the marrow [24]. This factor regulates iron metabolism, independently of iron stores, and responds to increased iron demands after stimulation of erythropoiesis by erythropoietin [25,26].

## 4. Pathogenesis of CRA

CRA is a cytokine-mediated disorder resulting from complex interactions between tumor cells and the immune system. Overexpression of certain inflammatory cytokines results in shortened survival of red blood cells, suppression of erythroid progenitor cells, impaired iron utilization and inadequate erythropoietin production [27,28]. Among inflammatory cytokines, macrophages play a crucial role because their activation is the main cause of increased erythrocyte destruction. In addition, other proinflammatory cytokines (e.g., IFN-γ, TNF-α, IL-1, and IL-6) make an important contribution to the etiopathogenesis of CRA. When inflammatory cytokine levels rise significantly, as in the case of chronic inflammatory diseases such as cancer, the number of destroyed erythrocytes also increases, and erythropoiesis is insufficient to compensate due to two main mechanisms: iron restriction and direct action of inhibitory cytokines on erythropoietic progenitors [29,30].

Examining the action of each cytokine in detail, IFN-γ inhibits erythropoiesis through a reduction in the number of erythroblasts [31] thus leading to lower erythroferrone levels.

IL-1 acts directly and selectively, suppressing replication and maturation of erythroid (BFU-e and CFU-e) precursors, reducing erythropoietin (EPO) receptor expression, and impairing EPO synthesis [32,33]. Moreover, IL-1 activates macrophages for erythrophagocytosis, thus inducing premature destruction and reduced survival of erythrocytes. It also causes several changes in energy metabolism and nutritional status, inducing anorexia [34], corticotropin-releasing factor and somatostatin secretion [35], with a consequent muscle mass reduction, which is typical of advanced cancer patients [36], and insulin synthesis, leading to hyperinsulinemia and insulin resistance [37]. These IL-1 mediated activities may concur with the onset of CRA in advanced cancer patients. In particular, erythropoiesis is negatively affected by low glucose availability and insulin resistance since commitment to the differentiation stages strictly depends on glucose metabolism [38].

IL-6, on the other hand, can interfere with iron homeostasis by impairing proliferation of erythroid progenitors and their response to EPO [39], and by altering liver gene expression and hepcidin synthesis [28]. An additional IL-6-mediated mechanism has been identified, namely Hb synthesis inhibition, which acts independently of the hepcidin-iron pathway as a consequence of impaired mitochondrial function (by decreasing membrane potential/oxidative phosphorylation) in maturing erythroid cells [40]. Moreover, IL-6 induces severe immune and metabolic alterations, such as cachexia, which characterize advanced cancer and contribute to CRA pathogenesis, and which can be prevented with anti-IL-6 monoclonal antibodies [41]. Moreover, it has also been shown that hepcidin stimulation might be due to cross-talk between the IL-6/JAK2-STAT3 pathway and a second inflammatory pathway through activin B, a bone morphogenetic protein receptor, and its Smad signaling mechanism [42,43].

More recently, IL-6 has been identified as the key determinant of muscle mass wasting in advanced cancer patients [44]. In particular, the activation of IL-6/STAT3 dependent regulation of the phosphatidylinositide 3-kinases/protein kinase B (PI3K/AkT) mammalian target of rapamycin (mTOR) pathway, together with increased degradation and low availability of amino acids, may contribute to defective erythropoiesis in advanced cancer patients. Indeed, red cell maturation and the synthesis of Hb depend on the activation of mTOR through an increased amino acid uptake [45,46]. It is clear that anorexia, associated with reduced food intake, and insulin resistance, with impaired glucose metabolism, also contribute to the inhibition of the mTOR pathway.

In turn, anemia, defined as a condition of reduced efficiency in the transport of O_2_ to peripheral tissues, may inhibit mTOR complex 1 (mTORC1) signaling, mainly as a consequence of impaired oxidative phosphorylation and reduced ATP synthesis, leading to mTOR inhibition.

TNF-α also has direct effects on hematopoiesis, impairing erythropoiesis and erythroid differentiation in vivo and in vitro, inducing increased immature erythroblast apoptosis, decreased mature erythroblast apoptosis, and reduced responsiveness of erythroid progenitors to EPO [47]. Furthermore, TNF-α induces metabolic changes in lipid metabolism, which are typical of advanced cancer patients, particularly those with cachexia [48]. In particular, TNF-α reduces the activity of lipoprotein lipase, which converts circulating triglycerides into free fatty acids (FFA), decreases the adipocyte expression of FFA transporters and the synthesis of enzymes participating in lipogenesis, and induces lipolysis [49]. TNF-α has also been shown to be involved in the onset of insulin resistance by increasing FFA levels, inhibiting the insulin receptor and insulin receptor substrate-1 (IRS-1) production and inducing IRS-1 Ser/Thr phosphorylation. TNF-α signaling negatively affects the peroxisome proliferator activated receptor gamma (PPAR-γ), which physiologically exerts a crucial regulatory action on lipid metabolism [50] since PPAR-γ downregulation leads to lipoatrophy [51] and reduces maturation of erythroid progenitors [52].

Interestingly, EPO deficit [53] partially explains the increase in ROS concentrations [3,6] correlated with chronic inflammation. Indeed, ROS (O°, H_2_O_2_, and OH-) can inhibit EPO synthesis, by mimicking a false O_2_ signal in renal peritubular interstitial cells. Oxidative stress can also increase erythrocyte fragility, decrease the amount of erythroid maturation, and reduce red cell survival [54,55,56,57]. ROS also mediate the inhibitory effect of proinflammatory cytokines on erythroid precursor proliferation [58]. In addition, an in vitro study demonstrated that sustained H_2_O_2_ levels induce liver hepcidin expression through STAT-3 phosphorylation by acting synergistically with IL-6, indicating another potential mechanism through which oxidative stress could contribute to CRA [59].

Although cancer patients may have several of the aforementioned contributing factors for anemia before their cancer diagnosis, CRA pathogenesis might also be a direct result of the cancer (anemia secondary to cancer [ASC]), its treatment (radiation), chemotherapy-induced anemia [CIA]), or chronic kidney disease (CKD). ASC is a direct result of the malignancy invading normal tissues, causing blood loss, marrow infiltration which inhibits the production of red cells, or inflammation, leading to functional iron deficiency. Myelosuppressive chemotherapy either alone or in combination with radiotherapy commonly contributes to the development of anemia and is referred to as CIA [60]. CKD, a result of renal injury from tumor invasion, chemotherapy, or age-related decline can be diagnosed in the majority of elderly cancer patients [61].

### Potential Role of Microbiota in Iron Metabolism

It has been shown that both intravascular and oral iron administration improve iron availability and ensure higher ferritin levels, often making such interventions the first choice in treating CRA. However, iron administration negatively affects both the alpha and beta bacterial diversity, leading to an increased risk of enteric pathogens [62].

Indeed, fecal sample analyses before and after iron therapy show a significant variation in bacterial composition, suggesting variable sensitivity among bacterial communities to the therapy [63]. This is due to the lack of intraluminal regulation of iron availability despite the existence of intravascular regulation. Therefore, when iron is exogenously provided, intestinal bacteria compete successfully for this essential nutrient [64]. Interestingly, many pathogens develop very efficient Fe^3+^ uptake mechanisms based on high-affinity Fe^3+^ chelators, so-called siderophores, to increase survival [64] and effective colonization [65].

Siderophores are small, high-affinity iron-chelating compounds secreted by bacteria, and they are the most prevalent means used by aerobic and facultative anaerobic bacteria families, such as Enterobacteriaceae, Streptomycetaceae, and Bacillaceae, to scavenge inorganic iron from the environment [62]. The mechanism of action, common to the different types of siderophores, is iron internalization by Fe^3+^ binding and Fe^3+^ presentation to the membrane receptors. The ferric-siderophore complex enters the bacterial cell where the mineral is reduced (Fe^2+^) becoming available for cellular metabolism. Meanwhile, the siderophore is released to be used again [65]. In order to gain a competitive advantage, many bacteria develop the capability to scavenge and use the siderophores of other species [64]. Iron availability in the intestinal lumen (in particular, the colon) is also a crucial trigger for virulence in a number of infections, such as malaria, HIV-1, and tuberculosis [66]. It has been shown, for instance, that increased luminal iron might upregulate the expression of proteins mediating iron efflux in *Salmonella typhimurium*, enterohemorrhagic *Escherichia coli* and in other organisms residing in macrophages, enhancing their growth and pathogenicity [62,67]. Highly pathogenic strains of *Yersinia enterolytica*, *Yersina pseudotuberculosis* and *Yersina pestis* have a highly pathogenic common insula (apparently acquired by horizontal transmission from another organism) which codes proteins necessary for synthesis, transport, and regulation of the “yersiniabactin” siderophore [66]. It is interesting to note that infection risk is increased by both iron deficiency and iron overload. In animal models, iron deficiency compromises various aspects of cellular immunity, but it is not clear how this extends to the deficiency levels commonly observed in humans [68]. The optimal level of host iron status may differ for different pathogens and different organs and cell types. Iron redistribution in the host against extracellular organisms may increase susceptibility to intracellular organisms [66]. Pathogens also manipulate the supply of cellular iron. For example, an initial liver Plasmodium infection regulates ferroportin and DMT-1 (divalent metal transporter 1) in hepatocytes to increase iron intake, but systemic infection stimulates hepcidin synthesis and exhausts hepatocyte iron storage, starving new sporozoites from a co-infecting plasmodium strain attempting to establish a secondary infection. This is an example of “interkingdom reporting” [66]. Moreover, iron can induce ROS formation in the gut, causing oxidative stress and consequently, intestinal epithelial damage. In turn, the host intestinal immune system responds with increased pro-inflammatory cytokines, intestinal barrier damage, decreased production of immunomodulatory compounds, and increased infection risk [62].

To date, many human studies have only been based on observed correlations, and further studies are needed to prove a causal relationship between iron–gut bacteria interactions and the development of gut inflammatory diseases, colorectal cancer, and CRA.

However, a worsening of the inflammation and symptomatology in inflammatory bowel disease patients and a change in the bacterial communities as a consequence of a high iron intraluminal presence have been reported [63]. At the same time, the host produces proteins, such as hemopexin, haptoglobin, transferrin, and lactoferrin, which bind the heme group to compete with siderophores and to decrease iron availability to levels that are insufficient for bacterial colonization and growth [65]. It is therefore clear that the presence of anemia in these conditions has a host defense function. In choosing “the lesser evil”, the body responds by upregulating blocking systems involved in the release of micronutrients (such as iron) essential to microbial proliferation and, consequently, related to chronic inflammation aggravation and the onset of comorbidities [69,70]. Hence, effective interventions must act upstream first resolving the infection and inflammation.

## 5. First-Line Therapy

CRA is associated with, and aggravates, the multiorgan failure that occurs in advanced cancer, compromising patients’ quality of life. Thus, CRA has a negative prognostic significance. Therapeutic strategies to treat CRA should target the multiple causes that trigger the disease and should include erythropoietic agents, iron supplementation or blood transfusions, nutritional supplementation, and anti-inflammatory therapies [27].

During erythropoiesis, homeostatic mechanisms compensate to increase EPO synthesis when Hb drops below 12 g/dl [71], and the greatest improvement in quality of life is achieved with Hb values of 12 g/dl, thus showing the relevance of keeping Hb values within this range [72]. 

Red blood cell (RBC) transfusions are a fast and effective therapeutic intervention to quickly improve the patient’s symptoms [60] by rapidly boosting Hb and hematocrit values [73]. In cases of severe symptomatic anemia or life-threatening anemia (Hb < 7–8 g/dl), RBC transfusions are particularly useful. According to the latest National Comprehensive Cancer Network guidelines [60], RBC transfusions should not be considered on the basis of a specific threshold value of Hb. Rather, they should be used in patients with symptomatic anemia, in high-risk patients (e.g., those undergoing high-dose chemo-or radiotherapy with the cumulative decrease of Hb levels) or asymptomatic patients with comorbidities, but not in multiple alloantibody patients. Several studies have shown a survival benefit in neoplastic patients receiving transfusions [74,75]. Moreover, blood transfusions are useful in treating the subjective symptoms of patients, such as breathlessness [76,77] and fatigue [76,77,78,79]. On the other hand, blood transfusions have significant acute and long-term risks, such as fever, allergic reactions, transmission of infectious diseases, alloimmunization, iron overload, and immunosuppression [80,81], and they have been correlated with a higher incidence of thromboembolic events and death in hospitalized neoplastic patients [82]. 

Recombinant human erythropoietin (rHuEPO, epoetin alfa) was approved by the Food and Drug Administration (FDA) in 1993 for the treatment of anemia in cancer patients. Currently, different short- and long-acting formulations of rHuEPO are available: r-HuEPOa, r-HuEPOb, and darbopoetin alpha. Thanks to its glucidic component, r-HuEPO has a longer half-life after subcutaneous administration than natural EPO, which has a half-life of 8.5 h (24 h for r-HuEPOa and 20.5 h for r-HuEPOb) [83].

More recently, several biosimilar EPOs have been developed and introduced in clinical practice: biosimilar epoetin alfa (e.g., BinocritR, Sandoz) and epoetin zeta (e.g., RetacritR, Hospira).

In a large Cochrane meta-analysis, which evaluated erythropoiesis-stimulating agents (ESAs) for the treatment of CRA in patients undergoing or not undergoing concomitant chemotherapy, rHuEPO was shown to achieve a significant reduction in RBC transfusions and a higher hematopoietic response thus improving quality of life, fatigue, and other specific anemia-related symptoms [84].

Several open-label nonrandomized community-based trials in advanced cancer patients with CRA have shown that a progressive amelioration in patients’ general health status obtained by ESAs was significantly correlated with increased Hb levels [85,86,87,88,89]. Furthermore, ESAs can also have neuroprotective, anti-inflammatory, vascular, and metabolic actions [90].

There is a growing body of evidence supporting the efficacy of intravenous iron administration, in combination or not with ESAs, in improving quality of life and decreasing the need for transfusion in cancer patients. Oral iron supplementation which is the first choice for treating anemia in patients with no inflammation, is inappropriate for treating inflammation-related anemia [27] due to inadequate intestinal absorption, metabolic disorders associated with inflammatory cytokines and gastrointestinal complications [91]. On the other hand, when intravenously administered, iron can be trapped directly by macrophages counteracting absorption problems. Saccharate iron ferric gluconate, like other less stable complexes, requires several low dose infusions, while more stable complexes, including ferric carboxymaltose, allow single infusions of high iron doses [91], which are reported to be well-tolerated with a low incidence of hypersensitivity reactions [92]. Currently there is no standard therapy, intravenous iron in combination or not with ESAs, or transfusion, is recommended [60]. Future strategies could include chelate-iron therapy, the use of hepcidin antagonists and cytokines or hormones that can modulate erythropoiesis under severe inflammatory conditions. In any case, further studies on anemic cancer patients are warranted.

## 6. Evidence of CRA Modulation by Lifestyle Interventions in Inflammatory Conditions

Although recent conventional therapies offer benefits to most patients, some remain anemic and therefore there is a need to develop a novel multitarget approach (Figure 1) to address persistent anemic conditions such as CRA. Moreover, considering the multifactorial mechanisms leading to CRA, mainly attributable to chronic inflammation, certain combined targeted approaches have already been shown to yield substantial benefits in terms of improving anemia and related symptoms. Recent evidence shows that in the absence of increased risk factors or comorbidities, CRA, its associated symptoms and impairment of the quality of life, can be improved by the administration of rHuEPO associated with the concomitant administration of low molecular weight heparin to counteract prothrombotic status, together with the appropriate supplements and adequate nutrition to support intestinal eubiosis [1]. The association of nutritional status with hemoglobin levels in cancer patients has not been sufficiently explored. A proper characterization of cancer patients with anemia on the basis of tumor staging and inflammation/metabolic-related symptoms is thus warranted to identify specific parameters that will enable the design and implementation of the optimal therapeutic strategy to treat CRA, in which inflammation and metabolic impairments seem to play crucial roles [93]. In this context, possible lifestyle interventions able to significantly modulate inflammatory conditions closely associated with CRA will be described.

### 6.1. Mediterranean Diet: Microbiota-Mediated Anti-Inflammatory Effects

Considering the etiopathogenesis of CRA, a diet with anti-inflammatory properties might help to improve the patient’s metabolic profile and increase antioxidant levels.

Intervening in the inflammatory milieu through the diet can also help to avoid malnutrition, which is at least partially attributable to a decrease in nutrient intake, but also closely linked to the effect of the inflammatory state on intermediary metabolism and on microbiota health [94]. The inflammatory response impacts nutrition by elevating resting energy expenditure and nitrogen excretion and, thereby, energy and protein requirements, respectively [95]. In particular, in the present review, we will examine how adherence to the Mediterranean diet may attenuate inflammation and oxidative stress [96].

Epidemiologic studies have shown that consumption of a Mediterranean diet is associated with a lower incidence of diseases related to low-grade systemic inflammation, such as metabolic, cardiovascular, and oncological pathologies [97,98,99].

As mentioned above, the cause of this chronic inflammation is the web of complex relationships between diet, the gastrointestinal microbiome, and the immune system, which leads to a state of metabolic endotoxemia defined as a 2- to 3-fold increase in circulating levels of bacterial lipopolysaccharide (LPS) [100]. These lipoglycans and endotoxins, found in the outer membrane of Gram-negative bacteria, are able to upregulate pro-inflammatory cytokine transcription through toll-like receptor (TLR) 4 activation [101].

As a primary source of LPS in the body, changes in gastrointestinal microbiota composition and/or production of microbial metabolites may alter the pool of pathogen-associated molecular patterns that encounter the intestinal epithelium and enter the bloodstream [102]. Increasingly, diet has been shown to be the primary mediator of gastrointestinal microbiota composition (alpha e beta diversity) and function [103].

For example, dietary components rich in bacteria that have an almost exclusively saccharolytic metabolism, such as Lactobacilli and Bifidobacteria, are considered potentially beneficial [104]. Both of these gut microbial species produce a variety of tryptophan catabolites, which, by decreasing intestinal permeability and, therefore, LPS translocation, are critical to the maintenance of intestinal homeostasis [105]. In addition, some of these catabolites enter the bloodstream and have anti-inflammatory and anti-oxidative effects [105]. Several studies have shown that daily intake of these probiotics, for example, through yogurt, kefir, and fermented vegetables, reduces the levels of pro-inflammatory cytokines and chronic inflammation biomarkers in premenopausal women [106,107]. In particular, kefir has been proposed to counteract hypertension, gastrointestinal and ischemic heart diseases, allergies [108], fatigue and related metabolites [109,110], and exercise-induced immune suppression [111].

Dietary fiber, whole-grain complex carbohydrates, legumes, and natural sugar alcohols, staples of the Mediterranean diet, are prebiotics that support healthy microbiota [112,113,114]. Degradation of whole-grain complex carbohydrates increases short-chain fatty acid (SCFA) production, improving immunity [115].

An observational study found a positive correlation between high-level adherence to the Mediterranean diet and SCFA fecal concentrations [116]. SCFAs are the end products of microbial fiber fermentation, mainly utilized by peripheral tissues (acetate), the liver (propionate), and the colonic mucosal cells (butyrate) [117].

In vivo studies have shown that prebiotic fiber supplementation increases butyrogenic bacteria species, such as *Faecalibacterium prausnitzii*, *Anaerostipes* spp., *Eubacterium* spp., and *Roseburia* spp., and bacteria cross-feeding with butyrogenic bacteria [118], such as Lactobacillus and Bifidobacteria, which do not produce butyrate but have the enzyme butyryl-CoA to convert other metabolites into butyrate [119].

Interestingly, microbial metabolites such as SCFAs have been amply investigated as critical cofactors and allosteric regulators of epigenetic processes influencing human health and disease, particularly cancer and response to treatment [120].

Promoted at the same biological level as prebiotics and probiotics, the beneficial effects of polyphenols on gut health have also been shown; thus, we have the concept of the “three P’s” [121]. Preclinical and clinical studies have reported that supplemental polyphenols and polyphenol-rich foods stimulate the growth of beneficial bacteria, such as Lactobacilli and Bifidobacteria [122], but also species, such as *Akkermansia* spp., *Faecalibacterium prausnitzii*, and *Roseburia* spp. [123]. These findings support the hypothesis that polyphenol-mediated anti-inflammatory effects are to be found in their microbiota modulating properties [124].

Notably, the low oral bioavailability of polyphenols may be increased by microbiota-mediated biotransformation [125] thus strengthening the bidirectional link between potential active compounds and the functional health of the gastrointestinal tract [126].

On the contrary, a diet rich in SFAs may alter the gastrointestinal microbiome and intestinal physiology in ways that contribute to metabolic dysfunction and systemic inflammation. Animal studies report that very high-fat diets are associated with a reduction in microbial diversity, a decreased Firmicutes:Bacteroidetes ratio, an increase in mucosal expression of lipid-related genes and LPS translocation [100,127]. Moreover, a diet rich in red meat, eggs, and dairy products, due to its high content of choline, lecithin, and carnitine, is a potential source of trimethylamine-N-oxide (TMAO). TMAO is formed from trimethylamine (TMA) and, in humans, a positive correlation between elevated plasma levels of TMAO and an increased risk for major adverse cardiovascular events and death has been reported [128,129]. The atherogenic effect of TMAO is attributed to alterations in cholesterol and bile acid metabolism, activation of inflammatory pathways and the promotion of foam cell formation.

Human and animal studies suggest that several families of bacteria are involved in TMA/TMAO production, namely, Deferribacteraceae, Anaeroplasmataceae, Prevotellaceae [130], and Enterobacteriaceae [131,132].

Moreover, large proteins in dairy products and red meat that are not completely digested can feed proteolytic bacteria, resulting in pro-inflammatory metabolite production [133,134,135,136]. The proper diet can, however, modulate the abundance of these species. For example, fiber consumption and enzymatic proteins, such as papain and bromelain (papaya, mango, and pineapple) help protein digestion and reduce colonic transit, resulting in an anti-inflammatory effect [137,138].

In order to reduce intestinal permeability and, therefore, systemic inflammation it is also important to reduce prolamins, alkaloids, saponins, and lectins, present in legumes, cereals, pseudocereals and vegetables belonging to the Solanaceae family (tomatoes, eggplants, and potatoes) [139,140]. It is also important to reduce the intake of highly refined flours, gluten [141], trans FA [142,143], and to lower pro-inflammatory omega 6 poly-unsaturated FA (PUFA-6) [144].

One of the most significant tests to detect the persistence of silent inflammation and one of the main biomarkers for cardiovascular and oncological diseases is the ratio between omega-6 and omega-3. Today, the FDA reports an average ratio of 25:1 in the ordinary diet, far from the ideal proportion of 4:1 or 2:1 [145,146].

Notably, dietary intake of PUFAs (i.e., body lipids of fatty fish, despite their choline content, the liver of white lean fish, nuts, and chia, flax, and canola seeds) is essential because, unlike saturated FA and mono-unsaturated FA (MUFA), they cannot be synthesized endogenously; hence, the definition of essential fatty acids (EFA).

The Mediterranean nutritional plan also promotes MUFA (olive oil, avocado, and sesame) consumption [147]. Extra virgin olive oil, with its more than 30 phenolic compounds, is one of the main staples of the Mediterranean diet and a primary source of added fat [148]. It confers its health benefits by blocking enzymatic steps of the pro-inflammatory eicosanoids signaling cascade [149].

Another beneficial action of the Mediterranean diet is linked to the high content of antioxidants and phytochemical vitamins in fruits and vegetables; hence, their apparent anti-inflammatory and microbiota modulating properties [114,150,151,152].

Regular consumption of ginger, green tea, garlic, black pepper, and curcuma, together with turmeric, is also reported to confer antioxidant and anti-inflammatory action [153,154,155,156].

In addition to the importance of specific dietary components, it is also crucial to choose cooking methods that do not reduce the nutritional value of food.

Over the past two decades a growing body of evidence has emerged pointing to the negative contribution of food-derived advanced glycation end products (AGEs) to the body’s pool of AGEs, with detrimental effects on oxidative stress and inflammation [157].

AGEs are a large and heterogeneous group of compounds originating from the spontaneous “Maillard reaction” between reducing sugars and free amino groups in amino acids. We know now that AGEs can also be generated by a variety of other reactions, including sugars, lipids, and amino acid oxidation, to create reactive aldehydes that covalently bind to proteins [158].

The main factors determining the AGE formation rate in food include nutrient composition (protein > fat > carbohydrate), temperature and duration of the heat application, humidity, pH, and metal traces [159]. Therefore, different cooking methods can substantially affect the potential AGE food content without necessarily changing the nutrient composition. In general, cooking at high temperatures and for a prolonged period generates the highest AGE content [160].

Notably, dietary AGEs may partially increase colon permeability and detrimentally modulate gut microbial ecology, adversely affecting host health [161].

The Mediterranean diet might be proposed as an anti-inflammatory and anti-oxidative therapeutic intervention, especially in light of its interrelationship with human gut microbiota, genobiome, and epigenome.

The human intestinal microbiome will pave the way, leading to a new frontier in human biology in which the human genome and bioactive products from the intestinal microbiome are closely linked to one another, forming an integral part of the “human metagenome”.

### 6.2. Supplementation to Reduce Chronic Inflammation and Potentially Counteract Anemia in Cancer Patients 

To increase the efficacy of CRA treatment, iron administration could be integrated with proper nutrition and also with specific molecules able to reduce chronic inflammation, the primary cause of this problem. There are many molecules with scientifically documented anti-inflammatory action, such as polyphenols, probiotics, vitamin D, PUFAs, and alpha-lipoic acid. Moreover, an open-label randomized prospective trial has shown similar efficacy for oral lactoferrin and for, i.e., iron, combined with rHuEPO-β, in CRA patients undergoing chemotherapy [162]. Any integration should be managed by professionals because there might be interaction between drugs and supplements that needs to be evaluated. In fact, an important but under-investigated problem is the potential interference of dietary supplements with drug absorption, transport, and/or metabolism. Many studies report interaction between supplements and drugs, in particular, many flavonoids could modulate the activities of CYP3A4, CYP2C9, and CYP1A2 and thereby interfere with the hepatic drug metabolism [163]. Fully understanding the pharmacokinetics and pharmacodynamics of drugs and supplements could be helpful when the two therapies are integrated. Clinical trials of drug–supplement interactions are the gold standard, but often they are only carried out when unexpected consumer side effects are reported. However, most of the clinical studies on supplements that were predicted to interact with drugs showed no clinically significant effects [164]. Often discrepancies between preclinical and clinical data concerning drug–botanical interactions are highlighted, as well as gaps in knowledge on potential absorption-, transport-, and metabolism-based interference. Moreover, problems resulting from the use of botanical dietary supplements are exacerbated by the lack of standardization of these products, patients under-reporting supplement use to their health care providers, and consumers delaying conventional medical care due to reliance on the supplements [165]. It is important to note that botanical dietary supplements are used in many different forms, such as teas, tinctures, pills, or salves. Even scientific studies on a specific botanical dietary supplement effect may differ based upon the species of plants used in the product, the sources of the botanicals, how the botanicals are prepared, how the product is formulated, and how the product is standardized [164]. Each of these variables can affect the biological effects of a botanical dietary supplement and the outcomes of a scientific study.

Nutritional supplements are widely used among cancer patients for their antineoplastic and antitoxic effects, but they must be prescribed consciously by well-informed professionals following a medical indication. Supplements with the strongest scientific evidence supporting their effectiveness are reviewed below (Table 1).

#### 6.2.1. Polyphenols

Polyphenols, a large group of compounds derived from plants, are generally used as supplements for their antioxidant, anti-inflammatory, anti-infective properties, and their modulation of gut microbiota composition and abundance; hence, they are also considered as prebiotics [124,152,166].

Polyphenols interfere with immune cell regulation, proinflammatory cytokine synthesis, and gene expression [167].

They can inactivate nuclear factor kappa light chain enhancer of activated B cells (NFkB), modulate mitogen-activated protein Kinase (MAPk) and the arachidonic acid pathway. Moreover, polyphenols inhibit PI3K/AkT, inhibitor of kappa kinase/c-Jun amino-terminal kinases (IKK/JNK), mTORC1, which is a protein complex that controls protein synthesis, and JAK/STAT, which suppresses TLRs and pro-inflammatory gene expression [167,168].

Polyphenols also inhibit high mobility group Box1 protein, an important chromatin protein that plays a key role in inflammation by interacting with nucleosomes, transcription factors, and histones regulating transcription [169].

These botanical supplements show an agonistic effect on aryl hydrocarbon receptors (AhRs) and bind xenobiotic-responsive elements in promoter regions of certain genes, including Foxp3 expression [170].

Polyphenols inhibit certain enzymes involved in ROS production, such as xanthine oxidase and NADPH oxidase (NOX), while they upregulate other endogenous antioxidant enzymes, such as superoxide dismutase (SOD), catalase, and glutathione (GSH) peroxidase (Px) [171].

Some studies have reported inhibitory effects on phospholipase A2 (PLA2), cyclooxygenase (COX) and lipoxygenase (LOX) leading to a reduction in prostaglandin (PG) and leukotriene (LT) production [166].

Finally, polyphenols exert their anti-inflammatory activities by directly affecting the count and differentiation of specific immune cells (i.e., oral administration of polyphenols increases T helper 1, regulatory T cells characterized by the (CD4 + CD25 + Foxp3+) phenotype, natural killer (NK) cells, macrophages, and dendritic cells (DCs) in Peyer’s patches and spleen) [172,173,174].

All the effects of polyphenols are associated with a wide range of health benefits for different chronic inflammatory diseases, including inflammatory anemia. In the following sections, some the most commonly used polyphenols will be described. 

#### 6.2.2. Curcumin 

Curcumin, one of the most popular supplements, is a natural polyphenolic compound isolated from turmeric (or Curcuma longa), a spice that has been widely used in traditional medicine for many centuries [175,176].

Curcumin reduces the expression of inflammatory cytokines (TNF-α and IL-1), adhesion molecules (intracellular adhesion molecule (ICAM-1) and vascular cell adhesion molecules (VCAM-1)) in human umbilical vein endothelial cells, and inflammatory mediators (prostaglandins and leukotrienes). It also inhibits some enzymes involved in inflammation (COX in mice, LOX in human endothelial cells, MAPK and IKK) [177]. Moreover, curcumin downregulates NFkB and STAT3, reduces TLR-2 expression and, in vivo, upregulates peroxisome proliferator-activated receptor gamma (PPAR-γ) in male adult rats [178,179].

This compound is known to exert anticancer effects by regulating a variety of intracellular signaling pathways and protein–protein interactions. Since one of the proposed mechanisms for hepcidin production involves intracellular and nuclear signal transduction after transmembrane receptor activation following some cytokine mediators such as IL-6, it has been noted that the covalent modification of STAT3 to its cysteine 259 residue by curcumin interrupts the phosphorylation, dimerization, and nuclear translocation of STAT3, thereby blocking hepcidin transcription. In a randomized double-blind, placebo-controlled study, it was shown that 6 grams of Curcuma, containing about 2% curcuminoids, induces a decrease in hepcidin levels (−19%), thereby increasing iron release [180].

One of the limitations of using Curcuma longa is the lack of bioavailability and, therefore, the consequent poor bioactivity of curcuminoids in vivo. There have also been reports of its potential chelating effect on metals, including iron [181]. To overcome those limitations, several preparations have been proposed (phytosomal, liposomal, nanoparticles, conjugation with various carriers, etc.). Promising results have been obtained using a formulation that combines liposomal technology with curcuminoids and cyclodextrin complexes [182].

#### 6.2.3. Resveratrol

Resveratrol, a polyphenol present in red wine grape, nuts, and other plant foods, has anti-inflammatory, anti-oxidant, and cardioprotective properties [183]. In vivo and in vitro studies demonstrate that resveratrol inhibits COX, inactivates PPAR-γ and induces endothelial nitric oxide synthase (eNOS) in murine and rat macrophages [167,184]. Similar inhibitory action on pro-inflammatory cytokines, such as TNF-α and IL-6, has been reported for a resveratrol analog, RVSA40, in murine macrophage cell lines. Resveratrol inhibits NFkB activation and VCAM-1 expression in LPS-stimulated human umbilical vein endothelial cells [185,186].

#### 6.2.4. Quercetin

Quercetin, present in apples, grapes, blueberries, green tea, onions, celery, and capers, is able to inhibit leukotriene biosynthesis in human polymorphonuclear leukocytes, to activate adiponectin production known for its anti-inflammatory effects, and to block nuclear translocation of p50 and p65 subunits of NFkB through inhibition of IkBα protein phosphorylation [167,187,188,189]. Moreover, it represses pro-inflammatory associated gene expression, NOS and COX-2, in murine macrophage cell line [190]. It can affect chromatin remodeling by blocking the action of histone acetyltransferase, and it inhibits the gene expression of TNF-induced IFN-γ-inducible protein 10 (IP-10) and macrophage inflammatory protein 2 (MIP-2). Lastly, it activates the production of adiponectin, known for its anti-inflammatory effects [191,192].

Quercetin blocks ERK, JNK phosphorylation in THP-1 activated human monocytes, while in murine macrophage cell lines it blocks phosphorylation and activation of JNK/SAPK (stress-activated protein kinases), ERK1/2 and p38, leading to a reduction in TNF-α transcription and expression [193].

#### 6.2.5. Epigallocatechin Gallate

Epigallocatechin gallate (ECGC) from green tea can lead to NFkB inactivation in human epithelial cells counteracting IKK activation and IkBα degradation. Moreover, it downregulates the expression of iNOS (inducible nitric oxide synthase), NO (nitric oxide) production in macrophages resulting in its immunomodulation [194,195,196].

In animal studies, epigallocatechin-3-gallate increases the number of functional Treg in spleens, pancreatic lymph nodes, and mesentheric lymph nodes. Similarly, in vitro treatment of Jurkat T cell ECGC boosts the expression of IL-10 and Foxp3, maintaining Treg identity [197].

EGCG reduces Th1 differentiation and Th17 and Th9 cell numbers in specific pathogen-free C57/BL6 female mice [198]. It reduces inflammation in various cell types by exerting an anti-MAPK activity, andit reduces IL-12 expression in LPS-activated murine macrophages by prohibiting p38 MAPK phosphorylation [199,200].

In LPS activated murine macrophages, green tea polyphenols not only suppress NFkB and MAPK pathways but also constrain the expression of COX-2 and the release of prostaglandin (PGE2) in murine macrophage cell lines [201,202].

#### 6.2.6. Vitamin D Supplementation

Recent literature shows an inverse correlation between vitamin D (assessed by serum 25-hydroxyvitamin D (25(OH)D)) and hepcidin concentration in patients with chronic inflammatory diseases [203,204,205]. Moreover, in newly diagnosed intestinal bowel disease in pediatric patients, as well as in mechanically ventilated critically ill adults and healthy controls, trials report that high-dose vitamin D administration significantly decreases serum hepcidin and CRP, suggesting a potential role for vitamin D in treating anemia [206,207,208]. An interesting in vitro study investigated the underlying mechanisms of the effects of vitamin D on the three key iron proteins: hepcidin, NRAMP1 (the endosomal iron transporter that transfers recycled iron from the late endosome to the cytosol) [209], and ferroportin. It was shown that the vitamin level was associated with decreased hepcidin, while it increased ferroportin and NRAMP1 mRNA expression in a dose-dependent manner in human macrophage-like monocytic cells in the presence of an inflammatory stimulus (i.e., exposure to LPS) [210]. 

It is known that vitamin D has potent anti-inflammatory properties reducing hepcidin stimulatory cytokines, such as IL-6 and IL-1β, and chemokine MCP-1 [211,212]. Reducing circulating IL-6 levels could lead to a reduction in hepcidin expression in liver hepatocytes, the major source of hepcidin, as well as in macrophages, major producers of inflammatory cytokines [213].

Taken together, these data suggest that vitamin D may have an important role in regulating cellular iron homeostasis via the hepcidin-ferroportin-NRAMP1 axis in macrophages to ameliorate iron availability during inflammation [214]. This regulation might be another key mechanism of vitamin D–mediated innate immune function, complementary to its reported effects on antibacterial proteins [215,216] and autophagy [217,218]. Although common to many chronic diseases [219], systemic inflammation and related anemia are likely to be particularly important in cancer patients, indicating disease severity [220] and correlating with the risk of developing multiple pathological complications, therapy responsiveness and the duration of disease-free survival [221].

#### 6.2.7. Lipoic Acid

Lipoic acid (LA) is an antioxidant able to produce its effects in aqueous or lipophilic environments. Lipoate is the conjugate base of LA, and the most prevalent form under physiological conditions. It presents a highly negative reduction potential, increases the expression of antioxidant enzymes, and participates in the recycling of vitamins C and E. Due to these properties, LA is called the “universal antioxidant” [222].

LA is also has anti-inflammatory action, which is independent of its antioxidant activity. LA has mainly been tested in cardiovascular diseases (CVD), obesity, pain, inflammatory diseases, and aging. It is well-defined as a therapy for preventing diabetic polyneuropathies, scavenging free radicals, chelating metals, and restoring intracellular glutathione levels which, otherwise, decline with age. Moreover, LA supplementation shows positive effects against cancer [222,223].

Though LA has long been touted as an antioxidant, it has also been shown to improve glucose and ascorbate handling, increase eNOS activity, activate Phase II detoxification via the transcription factor Nrf2, and lower matrix metallopeptidase-9 (MMP-9) and VCAM-1 expression through repression of NFkB [223].

LA and its reduced form, dihydrolipoic acid, may use their chemical properties as a redox couple to alter protein conformations by forming mixed disulfides. Beneficial effects are achieved with low micromolar levels of LA, suggesting that some of its therapeutic potential extends beyond the strict definition of an antioxidant [222,223].

LA has become a common ingredient in multivitamin formulas, anti-aging supplements, and even pet food. In light of its anti-inflammatory and antioxidant properties, it could be used to treat CRA [224].

#### 6.2.8. Lactoferrin

Lactoferrin (LF) is an iron-binding glycoprotein present in colostrum and many human biological fluids (milk, saliva, mucous secretions and tears); it exhibits anti-infective, anti-inflammatory, antioxidant [225], immune-modulating [226] and antineoplastic in vitro and in vivo [227] properties. It may be an interesting supplement for CRA treatment based on reports in the literature [162]; LF is also involved in the absorption of iron from dietary sources and it thus appears that supplementing LF could increase iron absorption [228]. LF seems to be a good integrative remedy for CRA because of its anti-inflammatory and antioxidant properties: it can inhibit the expression of inflammatory cytokines and promote the differentiation and growth of T lymphocytes, very important for the immune response in a cancerous disease. The anti-inflammatory property of LF is due to its ability to induce the expression of COX1 receptors in B lymphocytes and to increase IL 10 secretion with a reduction of INF-γ production by immune cells [229]. LF performs its functions in both innate and adaptive immunities, and it modulates the effects of inflammation both in mucosal and systemic immunities because the immunomodulatory properties of LF are due to its ability to interact with many cellular and molecular targets [230]. At the cellular level, it is involved in the modulation, migration, maturation, and various functions of immune cells as neutrophil granules. At the molecular level, it is able to bind iron and interact with many compounds, either soluble or cell-surface molecules: this capacity is a starting point for its extensive nearly ubiquitous immunoregulatory capacity [230]. An important role of LF in oxidation-reduction balance is based on its high antioxidant activity and inhibition of highly reactive oxidative agents. The ability of LF to bind the ferric ion (Fe^3+^) [231] is twice as high as transferrin, the main plasma protein whose function, as previously described, is transporting iron in the bloodstream; both are part of the same family of proteins, named “transferrins”, capable of binding and transferring ions Fe^3+^). Each LF molecule can bind two ferric ions to itself and based on this saturation it can exist in three distinct forms: apolactoferrin (iron-free), monoferric lactoferrin (linked to a single ferric ion), and ololactoferrin (which binds two ions to ferric itself) [231]. The oral integration of LF may be functional to maintain the oxide-reductive equilibrium, preventing iron from switching through Fenton reaction causing the synthesis of highly reactive oxidative compounds.

#### 6.2.9. Omega 3 Fatty Acids

Omega-3s (ω-3 PUFAs) include α-linolenic acid (ALA; 18:3 ω-3), stearidonic acid (SDA; 18:4 ω-3), eicosapentaenoic acid (EPA;20:5 ω-3), docosapentaenoic acid (DPA; 22:5 ω-3), and docosahexaenoic acid (DHA; 22:6 ω-3) [232].

Some of the oils containing FAs originate primarily from certain plant sources or are modified in plants, as well as marine, algal, and single-cell sources. Fish oils are sold as ω-3 PUFA supplements or in a concentrated form as ethyl esters (EEs) or acylglycerols, whereas algal, fungal, and single-cell oils have recently become popular as novel and renewable sources of long-chain ω-3 FAs. In addition, krill oil containing both triacylglycerol (TAG) and phospholipid (PL) forms containing EPA and DHA have been successfully marketed [233].

Researchers have also incorporated ω-3 PUFAs into different types of oils, such as borage oil and evening primrose oil to provide a better balance of PUFA components [234]. Interestingly, soy and other plants, such as flax and Brassica species, have been genetically modified to contain higher levels of ω-3 PUFAs. These novel and renewable sources of ω-3 offer oils without any fishy odor [235].

The bioavailability of ω-3 PUFAs is influenced by the form in which they exist, e.g., EE, TAG, or PL [236]. The superior bioavailability of TAGs compared to EEs has been confirmed by some recent findings, but this remains a controversial subject, as contrary findings have also been reported in recent years [237]. In addition, information on the relative bioavailability of PL forms of ω-3 PUFAs is limited and inconclusive [236,237,238].

Production of long-chain ω-3 PUFAs from ALA in the body is limited to rates of less than 4% at best; hence, their intake through the daily diet is important. According to the National Institute of Health [239], the required ALA level varies between 1.1 and 1.6 g/day depending on age and gender. In addition, they also recommend the intake of at least two servings of fish per week, thus providing nearly 0.3–0.45 g of EPA and DHA per day. The Food and Agricultural Organization (FAO 2010) of the United Nations recommends 0.5–0.6% ALA per day for the prevention of deficiency symptoms in adults, with a total ω-3 PUFA intake of 0.5–2% [232].

When the optimal lipid profile is not attained through diet intake, ω-3 integration should be considered, especially in the management of many serious diseases, such as cardiovascular disease (atrial fibrillation, atherosclerosis, thrombosis, inflammation, and sudden cardiac death, among others), diabetes, cancer, depression and various mental illnesses, age-related cognitive decline, periodontal disease, and rheumatoid arthritis [240].

In the literature, there are reports of the capability of ω-3 to partly inhibit many aspects of inflammation including leukocyte chemotaxis, adhesion molecule expression and leucocyte–endothelial adhesive interactions, prostaglandin and leukotriene production from the n-6 fatty acid arachidonic acid, and production of pro-inflammatory cytokines [241].

In addition, EPA gives rise to eicosanoids that often have lower biological potency than those produced by arachidonic acid, and EPA and DHA give rise to anti-inflammatory and inflammation resolving mediators called resolvins, protectins, and maresins [241].

Mechanisms underlying the anti-inflammatory actions of EPA and DHA include altered cell membrane phospholipid fatty acid composition, disruption of lipid rafts, modulation of the cell signaling cascade, inhibition of NFkB transcription factor and pro-inflammatory gene expression, and activation of the anti-inflammatory transcription factor PPAR γ.

Human trials demonstrate the benefit of oral and intravenous n-3 fatty acids in reducing inflammation in critically ill patients [242,243,244].

Lastly, ω-3 PUFAs may exert a positive action by changing microbiota composition and increasing the production of anti-inflammatory compounds such as SCFAs. In addition, a growing body of evidence from animal model studies indicates that the interplay between gut microbiota, ω-3 FAs and immunity helps to maintain intestinal wall integrity and interacts with host immune cells [245].

#### 6.2.10. Probiotics

Probiotics are defined as “non-pathogenic microorganisms that are able, once ingested in adequate quantities, to perform beneficial functions for the organism” [246]. To be defined as such, a probiotic should have the following characteristics [247,248,249]: being made up of live and human (species-specific) cells; resisting gastric acidity and bile; having muco-adhesiveness; being capable of colonizing the intestine; being able to modulate the immune system; producing antimicrobial substances and exercising antagonism against pathogens; not having antibiotic transmissible resistance (no plasmids) and being safe for use; having positive effects on human health; and being stable until the product expiration date [247]. Furthermore, based on the available scientific evidence, the minimum quantity sufficient to obtain temporary colonization of the intestine by a microbial strain is at least 10^9^ live cells per day. In recent years, there has been growing interest in probiotics, both as regards their clinical use and their use in scientific research. The microorganisms generally used as probiotics are Lactobacilli, Bifidobacteria and some yeasts. The clinical application of probiotics for the treatment of some pathologies dates back to the last century, with Nobel laureate Eli Metchnikoff, who first observed the positive effects of probiotics on human health, assuming that these effects derived from an improvement in the balance of intestinal microbiota through pathogenic bacteria inhibition [250]. Today, more and more studies confirm the benefits of using probiotics in many clinical settings [251,252,253,254], and the primary rationale for their use is to restore microbial balance. The microorganisms generally used as probiotics are Lactobacilli, Bifidobacteria, and some yeasts. Regular consumption of probiotics can modify intestinal epithelial cell proliferation, reduce permeability, restore redox homeostasis, and modulate low-grade inflammation after sustained exercise [255]. In particular, probiotic supplementation has been reported to increase the expression of mucin genes, enhancing mucin and antimicrobial peptide secretion [255], leading lactate-utilizing bacteria to produce butyrate [256], reducing zonulin, a marker of enhanced gut permeability, and modulating TNF- and exercise-induced protein oxidation [257]. Although more studies are needed to shed light on the underlying mechanisms that are involved [258], there are also reports describing the beneficial effects of gut microbiota health on mood disorders, such as anxiety and depression [259,260,261,262]. In addition, the gut microbiota is not only involved in inflammation, but is a “key orchestrator” of cancer therapy [263]. It modulates the activity, efficacy, and toxicity of several chemotherapy agents, such as gemcitabine, cyclophosphamide, irinotecan, cisplatin and 5-Fluorouracil, target therapy, and immunotherapy [264].

Hence, manipulating microbiota through probiotics, prebiotics, and diet, could be a strategy to improve the efficacy and mitigate the toxicity of anticancer drugs and to reduce inflammation and iron deficiency. Furthermore, in addition to their health benefits, the use of probiotics in cancer patients also seems to be safe [265,266,267].

Finally, many studies evaluate how probiotics can increase blood iron levels, iron bioavailability, and iron absorption [268]. In particular, the intake of *Lactobacillus Plantarum* 299 has been shown to have a clear role in increasing iron absorption from food sources [268,269,270].

Results from a systematic review and meta-analysis suggest that *L. Plantarum* 299 confers a beneficial effect on dietary non-heme iron absorption not only by promoting an anti-inflammatory immune response [271] that suppresses hepcidin [272], but also by promoting enterocyte iron uptake by enhancing mucin production at the intestinal surface [255], and through microbial metabolite production of p-hydroxyphenyllactic acid [273], a microbial by-product that can promote the reduction of ferric iron to the more bioavailable ferrous form [272].

**Table 1 nutrients-13-00482-t001:** Evidence of dietary supplementation counteracting anemia and inflammation.

Dietary Supplementation	Study Design (R/MA)	Effects			Major Targets
		Anti-Inflammatory	Antioxidant	Others	
Iron	R [274]			Replenishment of iron stores	1α-hydroxylase, IL-6, IL-1β, MCP-1
Vit. D	R/MA [275]	Yes		Stimulation of erythroid progenitor cell proliferation, reduction of hepcidin production	
Polyphenols	R [152]	Yes	Yes	Modulation of gut microbiota and mucosal integrity, anti-infective	NFkB, COX, NOX, SOD
Curcumin	MA [276]	Yes		Antineoplastic	TNF-α, IL-1β, COX, ICAM-1, VCAM-1, STAT3, NfKb, TLR, PPAR
Resveratrol	MA [277]	Yes	Yes	Cardioprotective	COX, PPAR, eNOS, NFkB, VCAM-1
Quercetin	MA [278]	Yes		Metabolic	NFkB, NOS, COX-2, TNF-α, adiponectin
Epigallocatechin gallate	R [193]	Yes	Yes	Immunomodulation	NFkB, iNOS, NO
Lipoic acid	MA [279]	Yes	Yes	Antineoplastic	VCAM-1, MMP-9 NFkB, eNOS, Nrf2
Lactoferrin	R [280]	Yes	Yes	Regulation of iron absorption, immunomodulation, anti-microbial, anti-viral, antineoplastic	COX-1, IL-10, INF-γ, TNF-α
Omega 3	MA [281]	Yes		Metabolic	NFkB, PPAR-γ, IL-6
Probiotics	MA [268]	Yes		restoration of microbial balance, immunomodulation, metabolic, promotion of enterocyte iron uptake	Hepcidin, p-hydroxyphenyllactic acid

Indicative and significant references proposed in the table were obtained from meta-analyses or, when not available, from review articles. The main mechanisms of each supplement are discussed in detail in paragraph 6.2. All supplements were administered chronically. Abbreviations: R = review; MA = meta-analysis; IL-6 = interleukin 6; IL-1β = interleukin 1 beta; MCP-1 = monocyte chemoattractant protein-1; NFkB = nuclear factor kappa beta; COX = cytochrome c oxidase; NOX = NADPH oxidase; SOD = superoxide dismutase; TNF-α = tumor necrosis factor alpha-like; ICAM-1 = intercellular adhesion molecule 1; VCAM-1 = vascular cell adhesion molecule 1; STAT3 = signal transducer and activator of transcription 3; TLR = toll-like receptor; PPAR = peroxisome proliferator activated receptor; eNOS = endothelial nitric oxide synthase; iNOS = inducible nitric oxide synthase; NOS = nitric oxide synthase; NO = nitric oxide; MMP-9 = matrix metallopeptidases 9; Nrf2 = nuclear factor erythroid 2–related factor 2; IL-10 = interleukin 10; INF-γ = interferon gamma.

## 7. Potential Role of Exercise in CRA

The inflammatory pathway can be seen as a potential therapeutic target for lifestyle interventions aiming to treat CRA not only through nutritional approaches but also through exercise. It has been observed that exercise protects against diseases associated with chronic low-grade systemic inflammation [282] and the importance of regular physical activity (PA) and exercise to prevent and treat chronic diseases is widely accepted. Therefore, what characteristics must exercise have to be safe and effective in lowering inflammation and how can exercise impact iron metabolism and erythropoiesis in CRA?

### 7.1. The Endocrine Function of the Skeletal Muscle and the Immunomodulatory Function of Exercise

Reviewing the benefits of exercise in treating or preventing chronic and degenerative conditions, a major role is attributable to its anti-inflammatory effect. Skeletal muscle has been identified as an endocrine organ that produces a variety of molecules, denominated “myokines”, able to control adaptive processes in skeletal muscle by acting as paracrine regulators of fuel oxidation, hypertrophy, angiogenesis, inflammatory processes, and extracellular matrix regulation. Based on the current state of knowledge, the major endocrine functions attributed to myokines are involved in body weight regulation, chronic low-grade inflammation modulation, insulin sensitivity amelioration, tumor growth suppression, and cognitive function improvement. Of particular interest is IL-6, defined as a “multitalented myokine” [283]. IL-6 represents the best-studied myokine and is a primary example of the auto-, para-, and endocrine effects of exercise-regulated myokines. IL-6 can act as a pro- or anti-inflammatory cytokine depending on its secretion mode and stimulus. After exercise-related stress, the circulating level of IL-6 increases drastically without the concurrent release of TNF-α and IL-1β. The massive IL-6 increase induces the expression of anti-inflammatory cytokines, such as IL-1ra, soluble TNF receptors, and IL-10 and inhibits the production of the proinflammatory cytokine TNF-α [284]. This anti-inflammatory response, despite concomitant mild exercise-induced inflammation, also allows the immune system to neutralize the pre-existent chronic low-grade inflammation that would not otherwise be counteracted. Obviously, this powerful reaction is not limited to the area of contracting muscle; rather it is systemic because the skeletal muscle, previously defined as a secretory organ, can directly modify the levels of circulating cytokines and act on distant organs and tissues through the circulatory system [285]. Although these massive modifications may be expected only after maximal or submaximal efforts, there are substantial increases in IL-6 also after moderate-intensity exercise [286], which is best suited for therapeutic purposes and may therefore play a fundamental role in counteracting CRA. In addition to directly reducing chronic low-grade inflammation, exercise can also fight it at the source, preventing or reducing visceral adipose tissue. Like muscle, adipose tissue is considered a secretory organ and is directly implicated in the secretion of cytokines [287]. Overweight and obesity dysregulate the production or secretion of these cytokines, favoring the release of pro-inflammatory molecules and becoming a major predisposing factor for chronic and degenerative diseases [288].

In addition to these mechanisms through which exercise can indirectly counteract CRA, exercise is also known to affect the circulating levels of hepcidin and to play a direct role in the development of disease. A variety of studies have shown that exercise induces notable physiological changes in the immune system [289] and that IL-6 itself is sufficient to induce hepcidin expression during inflammation [290].

Firstly, the fact that in post-exercise, hepcidin levels are found to be higher has raised concerns on the usability of exercise to prevent or counteract CRA. Examining all the evidence, increases in hepcidin after single bouts of exercise seem to be limited to 3–6 h of recovery, subsequently returning to physiological levels [291]. This acute change in circulating hepcidin is very likely associated with post-exercise IL-6 overexpression [292] and may therefore represent a transient upregulation that does not last over time. Furthermore, the increase in hepcidin is also related to exercise duration and intensity, with greater exertion associated with higher expression of the molecule [293]. The response to acute exercise, however, is not sufficient to account for what happens when the body undergoes repetitive exercise sessions, which may induce different responses and adaptations. Unfortunately, evidence of the effect of training on hepcidin production is fragmented, and results are often conflicting. Very often, circulating hepcidin increases after prolonged training periods, suggesting a trigger for anemia. However, many factors, such as gender, menstrual cycle, hormones, and baseline iron levels influence the hepcidin response to training [294]. Furthermore, exercise type, duration, and overall training load affect hepcidin release and, interestingly, the relationship between exercise load and hepcidin level seen for acute exercise was maintained for training [295]. Even though more studies assessing the effect of training load and exercise intensity on hepcidin and iron levels are needed, studies on humans and animals support the hypothesis that moderate-intensity aerobic exercise may be a safe method to improve iron status [296,297] by reducing inflammation and promoting erythropoiesis without significantly increasing circulating hepcidin.

### 7.2. Exercise Prescription: The Possible Role of Acute and Chronic Exercise

Myokines are involved in many exercise-induced adaptations and understanding the molecular pathways through which acute and chronic exercise can attenuate excessive inflammation, accelerate functional recovery, and muscle mass accretion following muscle damage is essential in developing therapeutic strategies.

The American College of Sports Medicine (ACSM) guidelines [298] and, more recently, a new Italian model both support the role of “exercise as therapy” in numerous chronic inflammatory diseases, as well as in cancer. Indeed, PA and exercise appear to offer a potential therapeutic approach to modulate low-grade inflammation. Nevertheless, it should be borne in mind that the possible anti-inflammatory effect varies according to the type of exercise. The ACSM guidelines for exercise prescription employ the so-called FITT-VP principle [298], which considers the frequency (F), intensity (I), time (T), and type (T) of exercise and its volume (V) and progression (P) over time in an individualized exercise training program. The application of the FITT-VP principle can influence various immune parameters, which are also associated with chronic inflammatory disease [284,299]. There is particular interest in the FITT-VP principle and its possible application to cancer treatment [300,301] This is especially important given the observation that different physiological responses can be associated with the immediate effects of a single bout of exercise (acute exercise or a single bout of exercise) or as a consequence of an acute exercise repeated over time (chronic exercise or training), and that these different responses are not uniform across all populations, suggesting certain individuals may be more susceptible to an excessive inflammatory response (i.e., patients with a chronic inflammatory disease compared to healthy subjects) [302,303]. Pedersen and Saltin [304] reviewed the literature concerning the effects of exercise as a therapy in several chronic diseases. Although specific exercise guidelines for numerous chronic diseases (i.e., cancer, diabetes mellitus, metabolic syndrome, etc.) are outlined, evidence-based exercise recommendations for every chronic inflammatory disease are still unclear, and the characteristics that an exercise protocol must have to be safe and effective in patients with multiple chronic diseases generating meta-inflammation are not adequately articulated. CRA therapeutic intervention is still controversial; however, treating the underlying chronic inflammation could be an effective strategy to restore or maintain iron homeostasis and modify hepcidin circulating expression in patients with CRA. To fill this gap, we investigated the evidence of the effects of acute and chronic exercise on systemic meta-inflammation and how different types of exercise can affect circulating levels of hepcidin and play a major role in the control of CRA.

### 7.3. Overview of Evidence of the Effects of Acute Exercise on the Immune Response in Patients with Chronic Inflammatory Diseases

A substantial number of studies in healthy and diseased subjects have demonstrated that markers of inflammation are reduced following a single bout of exercise [284,305,306], and it is likely that it is the anti-inflammatory effects of acute exercise in patients with chronic inflammatory disease to mediate the beneficial effect of exercise on health. A single bout of exercise induces alterations in the homeostasis of the whole body [307] and initiates a complex cascade of inflammatory events which depend on the type, intensity, duration, and familiarity of the exercise, as well as the age and clinical condition of the individual engaging in the exercise [307,308,309,310,311,312]. During acute exercise muscles release IL-6, and after low to moderate-intensity or intermittent exercise (over 2.5 h; e.g., marathon runners) [313] muscle mass and blood levels can increase significantly [284]. However, this increase in IL-6, occurring as soon as the exercise ends, disappears in a few hours after exercise (in the recovery phase) and is accompanied by increased levels of anti-inflammatory and cytokine inhibitors, such as IL-1rα and IL-10 [313,314,315]. Pro-inflammatory markers, such as TNF-α and IL-1β, do not seem to increase after acute moderate intensity exercise, although conflicting results have been documented [316]. It is therefore clear that acute bouts of exercise exert various effects on the immune system, which are typically transient in nature, while engaging in regular exercise (i.e., training) can reduce basal or resting levels of many inflammatory markers [317].

### 7.4. Overview of Evidence of the Effects of Chronic Exercise on the Immune Response in Patients with a Chronic Inflammatory Disease

When a single bout of exercise is repeated over time it becomes chronic exercise, which can also be defined as exercise training [318]. It appears that exercise training can attenuate or blunt the response of a single bout of exercise [319,320], and it is interesting to note how these responses may vary in relation to the exercise intensity, which represents an important parameter in response to exercise training. Engaging regularly in moderate-intensity exercise has been extensively recommended to counteract sustained meta-inflammation and can be considered as an immune system controller, able to improve defenses against infections and reduce the possibility of the onset of chronic diseases [321,322]; hence, it is associated with anti-inflammatory effects. On the other hand, studies have suggested that the effect of low-intensity training in healthy elderly persons can reduce baseline levels of pro-inflammatory markers, such as monocytes, CRP, and IL-6 [323,324]. Similarly, in healthy young adults, higher-intensity aerobic exercise training reduces stimulated production of TNF-α by monocytes [325] and affects resting levels of inflammatory markers during and after a period of intensive training [326,327].

However, the effects of exercise training on circulating inflammatory markers seem to depend not only on the intensity of the exercise, but also on the training status, age, and disorder involved [328], while baseline measurements of circulating inflammatory markers do not seem to differ greatly between healthy untrained and trained adults [299,326,329].

There are several mechanisms through which exercise training reduces chronic inflammation. In short, intense long exercise can lead, in general, to higher levels of inflammatory mediators, and thus might increase risk of injury and chronic inflammation. By contrast, moderate or vigorous exercise with appropriate resting periods can achieve maximum benefit.

### 7.5. Overview of Evidence of the Effects of Acute and Chronic Exercise on Serum Hepcidin Levels

Exercise can also affect hepcidin circulating levels and play a direct role in the iron-hepcidin homeostasis in humans [330,331,332,333]. Roecker et al. found a significant urinary hepcidin excretion increase 24 h post-competitive marathon (42.2 km). The excretion then declined from there, reaching baseline at 72 h post-exercise [334]. This increased urinary hepcidin excretion may have been induced by the well-documented increase of IL-6 and other circulating cytokines after marathon running.

Many studies have found an increase in post-exercise hepcidin levels hypothesizing that it might be caused by an increase in circulating cytokines. Analyzing the mechanisms of exercise-induced regulation of hepcidin, Kong et al. showed that hepcidin increases observed after single bouts of exercise seem to be limited to 3–6 hr of recovery and then return to physiological levels [291]. Contracting muscles contributes to most IL-6 production in the circulating response to exercise [313,335], and molecular IL-6 is very likely to be involved in this acute change in circulating hepcidin [292]. Furthermore, hepcidin has been presented as a crucial regulator of the iron absorption-degradation rate, which may be mediated by acute or chronic exercise and is also related to the duration and intensity of exercise. The literature reveals that a single bout of exercise (i.e., 30 min to 120 min of endurance exercise, intervallic or continuous), at a moderate or high-intensity (60% to 90% of VO_2peak_), facilitates the upregulation of hepcidin expression between 0 h and 6 h post-exercise, peaking 3 h after the end of the exercise session [293]. Even though more studies assessing the effect of single bout exercise or chronic exercise on hepcidin and iron levels are needed, the magnitude of the hepcidin response to exercise seems to be dependent on the pre-exercise status of iron (ferritin levels) and the circulating level of pro-inflammatory cytokines (particularly, IL-6). Moreover, the intrinsic characteristics of exercise (i.e., FITT-VP principle) seem to play a crucial role in the release of hepcidin.

### 7.6. Overview of Evidence of the Effects of Acute and Chronic Exercise on Microbiota Diversity

As mentioned above, oral iron supplementation plays a fundamental role in replenishing iron storage, and it is the first line of therapy to treat iron deficiency. However, oral supplementation could negatively affect microbiota diversity, causing local inflammation, and may not be effective if the ingested iron cannot be absorbed. Hence, in order to obtain positive results from oral iron supplementation therapies, it is fundamental to have or maintain healthy microbiota. It is now accepted that reduced gut microbial diversity is linked to the development of chronic diseases, both locally and systemically, and that a high level of biodiversity, on the contrary, is, typical of healthy, long-living subjects [336]. Recent evidence has shown that exercise increases this diversity primarily by reducing inflammation. Moreover, exercise-induced enhancement of the immune system, weight loss, and faster digestion are further reasons for proposing exercise to enhance gut microbial diversity and treat CRA [337]. Several diseases cause disrupted signaling in the host physiology, and exercise has been shown to enhance host-microbe interactions, but the extent to which exercise favorably alters innate immunity is only partially understood. Regarding exercise intensity, evidence suggests that moderate PA is associated with an anti-inflammatory status in the intestinal lymphocytes post-exercise (TNF-α expression is suppressed while IL-10 is increased) [338]; it tempers intestinal barrier dysfunction; it preserves mucous thickness and intestinal permeability [339]. Diminished bacterial translocation and upregulated anti-microbial protein production have also been associated with moderate exercise [339]. In particular, resistance exercise ameliorates the pro-inflammatory status of elderly subjects through attenuation of the TLR2 and TLR4 signaling pathways [340]. By contrast, strenuous, prolonged endurance exercise may reduce gastrointestinal blood flow [341], leading to hypo-perfusion [342,343], susceptibility to endotoxins (“leaky gut”, endotoxemia) [334], and an overall increased expression of pro-inflammatory modulators (TNF-α, IL-1, IL-6, IL-1ra) [289,338,344]. In addition, excessively intense exercise may compromise the mesenteric redox environment, therefore weakening the activity of antioxidant enzymes [345].

PA might be an important component of a multi-target integrated approach to treating CRA due to its health-promoting effects, but more research is needed to establish the optimal breaking point for exercise duration, intensity, and frequency, so that optimal personalized protocols can be developed.

### 7.7. Overview of the Role and Efficacy of Exercise Interventions in Patients with Cancer

Over the past two decades, a growing body of evidence has underscored the association between physical inactivity and the risk of developing various types of tumors [346], while PA has been shown to help prevent several types of cancer, including breast, colon, endometrial, kidney, bladder, esophageal, and stomach. Indeed, large epidemiological studies have shown that regular and moderate to vigorous exercise can reduce cancer incidence by 40% [347]. In addition, exercise oncology is generally acknowledged to be associated with positive changes in physiological measures (e.g., cardiopulmonary fitness, physical function, and body composition), as well as in patient-reported outcomes (e.g., fatigue, sleep quality, and sense of empowerment) [348,349]. These parameters are of direct significance for the amelioration of cancer prognoses, but emerging evidence shows that exercise is also directly linked to the control of tumor biology, and thus may ultimately improve clinical outcomes. Recent studies have demonstrated that a specific and individualized exercise program controls cancer progression through direct effects on tumor intrinsic factors (e.g., growth rate, metastasis, tumor metabolism, and immunogenicity of the tumor), regulates tumor growth through interplay with systemic factors, alleviates adverse events related to cancer and improves cancer treatment efficacy [350]. All the above-mentioned benefits are the result of both acute and chronic exercise administration, with cumulative benefits manifesting themselves over time. Acute sessions have a dual function (physical and endocrine), increasing blood flow, shear stress on the vascular bed, temperature, and sympathetic activation while inducing the release of catecholamines, myokines and exercise-induced hormones. Both functions result in a cancer preventive effect by dampening the processes involved in the promotion and progression of malignancy. On the other hand, chronic training adaptations lead to improved cytotoxic immune function and reduce systemic inflammation [350]. In addition, acute and chronic physiological changes in response to PA inhibit the proliferation of cancer cells [351,352] and reduce their tumorigenic potential [302,303].

Evidence that PA may play an important role in cancer prevention and treatment efforts has grown rapidly as new epidemiological data on this topic has piled up over the past decade. 

The ACSM released updated guidance and recommendations on the role of PA and exercise in cancer prevention and survivorship and recommends that cancer survivors should avoid inactivity and engage in 150 min per week of moderate-intensity aerobic exercise at least three times per week for a minimum 30 min. 

Strength (resistance) training should be performed at least two days per week and should involve the major muscle groups. In addition, the guidelines emphasize the importance of adapting exercise regimens to individual abilities, while also taking into consideration the effects of surgery as well as the side effects of chemotherapy, immunotherapy, or radiation therapy [353]. Although the evidence supporting the role of exercise in counteracting the development of tumors is strong and consistent, the literature remains inadequate in terms of providing sufficient information for the development of specific prescriptions according to cancer type, timing of treatment, and/or types of treatment or FITT components.

## 8. Future Direction: A Multitarget Approach and the Age of COVID-19

A multidisciplinary approach to treat CRA in cancer patients is essential to achieve optimal long-term outcomes. The novel 2019 coronavirus (COVID-19) pandemic is a global public health emergency that has altered routine cancer care, including allied health and supportive care interventions. Various groups have prepared guidance documents to assist oncologists and hematologists to adapt their clinical practice to facilitate patient-centered cancer care in this new environment [354,355]. However, little attention is being given to the concrete actions through which the patient can, even in times like these, reduce, prevent, or nullify related processing toxicity, mitigate cancer-related fatigue, restore physical function, exercise capacity, and maintain proper nutrition.

Based on available data, cancer patients have a twofold increased risk of COVID-19 infection and a higher morbidity and mortality rate than the general population because of their systemic immunosuppressive state caused by the malignancy and anticancer treatments (surgery/chemotherapy, target therapies, immunotherapy, and radiation) [356,357]. Therefore, the multitarget approach, which may bolster the immune system of cancer patients, could also help support the healthcare system in these uncertain and difficult times that we are facing during the COVID-19 pandemic.

## 9. Conclusions

In recent years, more and more data on the pathophysiology of CRA has been collected, and new evidence on integrated approaches to treat CRA is emerging. However, a successful comprehensive clinical approach based on multitarget strategies and lifestyle interventions to treat inflammatory CRA is still lacking. Currently, oral iron administration is the main conventional therapy for CRA even though it has been shown to impair the gut microbiota profile and cause serious gastrointestinal problems, such as diarrhea and morbidity. Moreover, this strategy does not always induce a significant amelioration of the patient’s iron status, particularly in cases when the anemia is not related to iron deficiency per se. On the contrary, evidence shows that a personalized nutritional plan, based on the Mediterranean diet, properly prescribed functional supplements, and regular physical activity at moderate or low intensity, can significantly improve patients’ general health status, promote microbiota diversity, and positively impact their metabolic profile and immune system. The above-mentioned treatment plan has been shown to reduce oxidative stress and inflammation, which are the underlying mechanisms in CRA. More studies are needed to explore the applicability of the integrated approach and validate a working protocol in CRA.

In conclusion, the patient’s physiological status should be objectively assessed, and the clinician, nutritionist and exercise specialist should work as a multidisciplinary team tailoring interventions to the anemia etiopathogenesis and patient’s overall clinical state.

## Figures and Tables

**Figure 1 nutrients-13-00482-f001:**
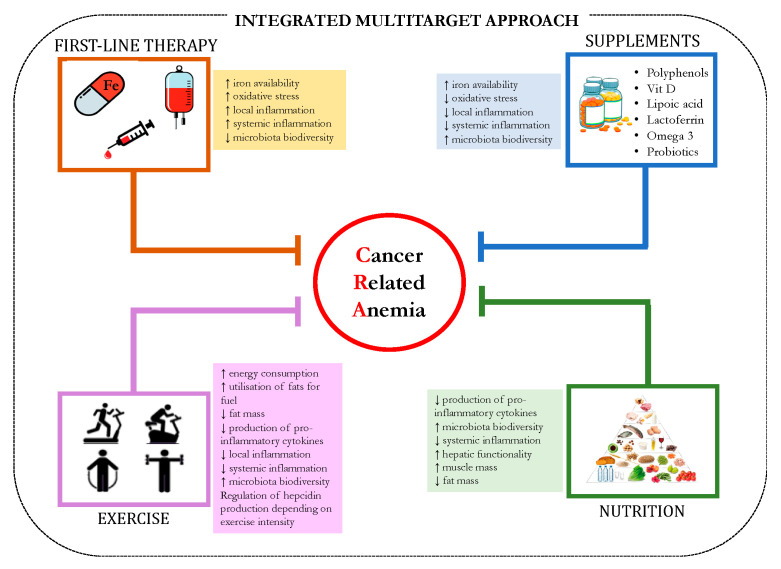
An overview of the multitarget approach to address cancer related anemia (CRA).

## Data Availability

Data sharing not applicable.

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
