# Peer review of "Cancer Related Anemia: An Integrated Multitarget Approach and Lifestyle Interventions"

_nutrients, 2021, doi:10.3390/nu13020482_

Round 1
Reviewer 1 Report
This is an interesting review, speculating on the role of nutrition and exercise in management of cancer-related anemia. The mechanistic discussion is excellent. The speculative discussion is responsible, distinguishing potential benefits through modulation of inflammatory stimuli to higher order benefits on hepcidin metabolism where there are data to support them.
There are some important errors that need attention:
- Line 777: I assume you mean 10[superscript]9 cells not 109.
- Line 1050: You state that the standard of care is oral iron. As a physician with a particular research and clinical interest in this topic, I am not aware of any authoritative guidelines that take that position. A better case could be made for intravenous iron, although that position also has a lot of caveats. I think a more accurate comment would be that, other than EPO with or without parenteral iron, or transfusion, there is no standard therapy.
There is some awkwardness in English usage and some errors. The manuscript would benefit from a review for that specifically. For example, on line 104, "maintained" would be more correct than "stabilized". The opening phrase "Interesting to note..." (and variations of it) is used a lot. It would be more correct to say "It is interesting to note...". On line 781, it should say "...Nobel Prize recipient Elie Metchnikoff..."
As a very minor point, the page numbering seems to reset after Table 1.
Author Response
Most of the revisions prompted by the reviewers’ comments are minor and require no further explanation than what appears in the responses below:
Reviewer 1
This is an interesting review, speculating on the role of nutrition and exercise in management of cancer-related anemia. The mechanistic discussion is excellent. The speculative discussion is responsible, distinguishing potential benefits through modulation of inflammatory stimuli to higher order benefits on hepcidin metabolism where there are data to support them.
There are some important errors that need attention:
R1. 1. Line 777: I assume you mean 10[superscript]9 cells not 109.
Authors: we agree and we modified it as 10[superscript]9
R1. 2. Line 1050: You state that the standard of care is oral iron. As a physician with a particular research and clinical interest in this topic, I am not aware of any authoritative guidelines that take that position. A better case could be made for intravenous iron, although that position also has a lot of caveats. I think a more accurate comment would be that, other than EPO with or without parenteral iron, or transfusion, there is no standard therapy.
Authors: we agree with the Reviewer, we modified the test as suggested, see the marked changes, we also added a new reference: Rodgers, G. M. The role of intravenous iron in the treatment of anemia associated with cancer and chemotherapy. Acta Haematologica 2019 [60]:
There is a growing body of evidence supporting the efficacy of intravenous iron administration, in combination or not with ESAs, in improving quality of life and decreasing the need for transfusion in cancer patients. Oral iron supplementation which is the first choice for treating anemia in patients with no inflammation, is inappropriate for treating inflammation-related anemia [27] due to inadequate intestinal absorption, metabolic disorders associated with inflammatory cytokines and gastrointestinal complications [91]. On the other hand, when intravenously administered, iron can be trapped directly by macrophages counteracting absorption problems. Saccharate iron ferric gluconate, like other less stable complexes, requires several low dose infusions, while more stable complexes, including ferric carboxymaltose, allow single infusions of high iron doses [91], which are reported to be well-tolerated with a low incidence of hypersensitivity reactions [92]. Currently there is no standard therapy, intravenous iron in combination or not with ESAs, or transfusion, is recommended [60]. Future strategies could include chelate-iron therapy, the use of hepcidin antagonists and cytokines or hormones that can modulate erythropoiesis under severe inflammatory conditions. In any case, further studies on anemic cancer patients are warranted.
R1. 3. There is some awkwardness in English usage and some errors. The manuscript would benefit from a review for that specifically. For example, on line 104, "maintained" would be more correct than "stabilized". The opening phrase "Interesting to note..." (and variations of it) is used a lot. It would be more correct to say "It is interesting to note...". On line 781, it should say "...Nobel Prize recipient Elie Metchnikoff..."
Authors: The manuscript has been careful revised by a native English speaker to improve the readability.
R1. 4. As a very minor point, the page numbering seems to reset after Table 1.
Authors: We modified it.
Reviewer 2 Report
In this review the authors explore the cancer related anemia (CRA) and and integrated multitarget approach and lifestyle based interventions.
The manuscript is very interesting, complete and well written.
There are no fundamental or preclusive concerns or limitations.
There are same points or revisions that should be considered:
- The review describes CRA pathophysiology. The role of cytokines and enzymes involved in iron metabolism are well described. The potential role of supplements, nutrition and probiotic on CRA pathophysiology are assumed on their proven effects in other setting, but real data of these interventions in CRA are not described. I think a specific section that review the efficacy of these interventions in CRA should be included.
- In the same way, data about efficacy of exercise in CRA or cancer should be included.
Author Response
Reviewer 2
Comments and Suggestions for Authors
In this review the authors explore the cancer related anemia (CRA) and integrated multitarget approach and lifestyle-based interventions.
The manuscript is very interesting, complete and well written. There are no fundamental or preclusive concerns or limitations.
There are same points or revisions that should be considered:
R.2.1. The review describes CRA pathophysiology. The role of cytokines and enzymes involved in iron metabolism are well described. The potential role of supplements, nutrition and probiotic on CRA pathophysiology are assumed on their proven effects in other settings, but real data of these interventions in CRA are not described.
I think a specific section that review the efficacy of these interventions in CRA should be included.
Authors: Currently, there is one clinical trial where a supplement, i.e. lactoferrin, has been used in CRA together with rHuEPO, showing similar effect as i.v. iron administration [227]. All the supplements taken into consideration in this review have potential reasoning of use thanks to their anti-inflammatory, anti-oxidative and eubiotic effects, counteracting the underlying chronic inflammation in CRA. We have rephrased accordingly the title of paragraph 6.2 and, in the conclusion, we have now highlighted the need for further studies to specifically validate the efficacy of these supplements in CRA integrated therapy.
- In the same way, data about efficacy of exercise in CRA or cancer should be included.
Authors: we agree with the Review consideration and we add a new paragraph (see 7.7)
7.7. Overview of the role and efficacy of exercise interventions in patients with cancer
Over the past two decades, a growing body of evidence has underscored the association between physical inactivity and the risk of developing various types of tumors [351], while PA has been shown to help prevent several types of cancer, including breast, colon, endometrial, kidney, bladder, esophageal, and stomach. Indeed, large epidemiological studies have shown that regular and moderate to vigorous exercise can reduce cancer incidence by 40% [352]. In addition, exercise oncology is generally acknowledged to be associated with positive changes in physiological measures (e.g., cardiopulmonary fitness, physical function, and body composition), as well as in patient-reported outcomes (e.g., fatigue, sleep quality, and sense of empowerment) [353; 354]. These parameters are of direct significance for the amelioration of cancer prognoses, but emerging evidence shows that exercise is also directly linked to the control of tumor biology and thus may ultimately improve clinical outcomes. Recent studies have demonstrated that a specific and individualized exercise program controls cancer progression through direct effects on tumor intrinsic factors (e.g., growth rate, metastasis, tumor metabolism, and immunogenicity of the tumor), regulates tumor growth through interplay with systemic factors, alleviates adverse events related to cancer and improves cancer treatment efficacy [355]. All the above-mentioned benefits are the result of both acute and chronic exercise administration, with cumulative benefits manifesting themselves over time. Acute sessions have a dual function (physical and endocrine), increasing blood flow, shear stress on the vascular bed, temperature, and sympathetic activation while inducing the release of catecholamines, myokines and exercise-induced hormones. Both functions result in a cancer preventive effect by dampening the processes involved in the promotion and progression of malignancy. On the other hand, chronic training adaptations lead to improved cytotoxic immune function and reduce systemic inflammation [355]. In addition, acute and chronic physiological changes in response to PA inhibit the proliferation of cancer cells [356; 357] and reduce their tumorigenic potential [358; 359].
Evidence that PA may play an important role in cancer prevention and treatment efforts has grown rapidly as new epidemiological data on this topic has piled up over the past decade.
The ACSM released updated guidance and recommendations on the role of PA and exercise in cancer prevention and survivorship and recommends that cancer survivors should avoid inactivity and engage in 150 minutes per week of moderate-intensity aerobic exercise at least three times per week for a minimum 30 minutes.
Strength (resistance) training should be performed at least two days per week and should involve the major muscle groups. In addition, the guidelines emphasize the importance of adapting exercise regimens to individual abilities, while also taking into consideration the effects of surgery as well as the side effects of chemotherapy, immunotherapy, or radiation therapy [360]. Although the evidence supporting the role of exercise in counteracting the development of tumors is strong and consistent, the literature remains inadequate in terms of providing sufficient information for the development of specific prescriptions according to cancer type, timing of treatment, and/or types of treatment or FITT components.
